# Colocalization of BRCA1 with Tau Aggregates in Human Tauopathies

**DOI:** 10.3390/brainsci10010007

**Published:** 2019-12-20

**Authors:** Masanori Kurihara, Tatsuo Mano, Yuko Saito, Shigeo Murayama, Tatsushi Toda, Atsushi Iwata

**Affiliations:** 1Department of Neurology, Graduate School of Medicine, The University of Tokyo, Tokyo 113-8655, Japan; mkurihara-tky@umin.ac.jp (M.K.);; 2Research Fellow, Japan Society for the Promotion of Science, Tokyo 102-0083, Japan; 3Department of Pathology and Laboratory Medicine, National Center Hospital, National Center of Neurology and Psychiatry, Tokyo 187-8551, Japan; 4Department of Neurology and Neuropathology (the Brain Bank for Aging Research), Tokyo Metropolitan Geriatric Hospital and Institute of Gerontology, Tokyo 173-0015, Japan

**Keywords:** BRCA1, tau, protein aggregation, DNA repair, Pick’s disease, progressive supranuclear palsy, corticobasal degeneration, Alzheimer’s disease

## Abstract

The mechanism of neuronal dysfunction via tau aggregation in tauopathy patients is controversial. In Alzheimer’s disease (AD), we previously reported mislocalization of the DNA repair nuclear protein BRCA1, its coaggregation with tau, and the possible importance of the subsequent DNA repair dysfunction. However, whether this dysfunction in BRCA1 also occurs in other tauopathies is unknown. The aim of this study was to evaluate whether BRCA1 colocalizes with tau aggregates in the cytoplasm in the brains of the patients with tauopathy. We evaluated four AD, two Pick’s disease (PiD), three progressive supranuclear palsy (PSP), three corticobasal degeneration (CBD), four normal control, and four disease control autopsy brains. Immunohistochemistry was performed using antibodies against BRCA1 and phosphorylated tau (AT8). Colocalization was confirmed by immunofluorescence double staining. Colocalization of BRCA1 with tau aggregates was observed in neurofibrillary tangles and neuropil threads in AD, pick bodies in PiD, and globose neurofibrillary tangles and glial coiled bodies in PSP. However, only partial colocalization was observed in tuft-shaped astrocytes in PSP, and no colocalization was observed in CBD. Mislocalization of BRCA1 was not observed in disease controls. BRCA1 was mislocalized to the cytoplasm and colocalized with tau aggregates in not only AD but also in PiD and PSP. Mislocalization of BRCA1 by tau aggregates may be involved in the pathogenesis of PiD and PSP.

## 1. Introduction

Tauopathies, including Alzheimer’s disease (AD), Pick’s disease (PiD), progressive supranuclear palsy (PSP), and corticobasal degeneration (CBD), are neurodegenerative disorders characterized by aggregated tau protein deposition in the brain. Each tauopathy shows characteristic tau deposition either composed of the three-repeat (3R) isoform of tau, four-repeat (4R) isoform of tau, or both [1]. The extent of tau deposition correlates with neuronal loss and clinical symptoms and is thus considered the major cause of these diseases. The mechanism of neuronal dysfunction and neuronal loss via tau aggregation remains controversial.

In AD, we previously reported that tau may cause neuronal dysfunction by coaggregation with BRCA1. BRCA1 is encoded by the *BRCA1* gene, which is known to cause hereditary breast and ovarian cancer syndrome [2] and Fanconi anemia [3,4]. BRCA1 performs various functions, including DNA damage signaling, DNA repair, cell cycle regulation, protein ubiquitination, chromatin remodeling, transcriptional regulation, mRNA splicing, and apoptosis, to maintain genomic integrity in the nucleus [5,6]. In AD, although DNA damage induced by extracellular amyloid β (Aβ) was accompanied by upregulation of BRCA1 protein in neurons [7], BRCA1 colocalized with tau aggregates in the cytoplasm and showed strong immunostaining as previously described [8], and was insoluble in a tau dependent manner [7]. Upregulation and cytosolic localization of BRCA1 was also reported later in human fibroblasts and neurons derived from induced pluripotent stem cells [9]. In AD mouse models, knockdown of BRCA1 in the hippocampus in the presence of extracellular Aβ increased the most toxic form of DNA damage, DNA double-strand breaks, and impaired the neuronal morphology, electrophysiological characteristics, and cognitive function in mice [7,10]. Thus, the coaggregation of BRCA1 with tau observed in advanced AD patients and reduced functional BRCA1 in the nucleus can lead to insufficient DNA repair and neuronal dysfunction.

However, whether colocalization and coaggregation of BRCA1 with tau also occurs in other human tauopathies had been unknown until recently [11]. The aim of this study was to evaluate whether BRCA1 colocalizes with tau aggregates in the cytoplasm in human tauopathy patients’ brains.

## 2. Materials and Methods

### 2.1. Subjects

Brain specimens of patients with histopathologically confirmed AD (*n* = 4), PiD (*n* = 2), PSP (*n* = 3), CBD (*n* = 3), dementia with Lewy bodies (DLB; *n* = 1), Parkinson’s disease with dementia (PDD; *n* = 1), and normal controls (NC; *n* = 4) were obtained from the Brain Bank for Aging Research at the Tokyo Metropolitan Geriatric Hospital and Institute of Gerontology in Tokyo, Japan, and brains of patients with multiple system atrophy cerebellar type (MSA-C; *n* = 1) and amyotrophic lateral sclerosis (ALS; *n* = 1) were obtained from the University of Tokyo Hospital in Tokyo, Japan. This research received approval from the ethics committee of the University of Tokyo (approval G2183–20). The presence of 3R tau isoform in PiD and 4R tau isoform in PSP and CBD brains [1] was confirmed by immunohistochemistry studies using primary antibodies against the 3-repeat isoform RD3 (8E6/C11; Merck Millipore, Darmstadt, Germany; Cat# 05-803) and 4-repeat isoform RD4 (1E1/A6; Merck Millipore; Cat# 05-804). Table 1 shows the clinical and neuropathological characteristics of the patients, including Braak stages of Aβ depositions and neurofibrillary tangles (NFT) [12] and the Brain Bank for Aging Research staging for Lewy body pathology [13,14]. After systematic neuropathological assessment, brain regions containing characteristic protein inclusions, namely, the medial temporal lobe for AD, NC, DLB, and PDD, frontal lobe for PiD and MSA-C, precentral gyrus (frontal lobe) and midbrain for PSP and CBD, and precentral gyrus (frontal lobe) for ALS, were used for further assessment.

### 2.2. Neuropathological Examination

The autopsy specimens were fixed with formalin and embedded in paraffin according to standard procedures. Sections were de-paraffinized, rehydrated, and subjected to an antigen retrieval procedure in sodium citrate buffer containing 10 mM sodium citrate pH 6.0 and 0.05% Tween 20 for 20 min at 110 °C.

For immunohistochemical studies, the sections were incubated with 10% normal goat serum for 30 min at room temperature to block non-specific binding and were immunostained with mouse monoclonal primary antibodies against phosphorylated tau (p-tau) (AT8; Thermo Fisher Scientific, Waltham, MA, USA; Cat# MN1020; 1:1000), BRCA1 (MS110; Abcam, Cambridge, UK; Cat# ab16780; 1:1000), phosphorylated α-synuclein (psyn#64; Fujifilm Wako Pure Chemical Co., Osaka, Japan; Cat# 015-25191; 1:30,000), phosphorylated TDP-43 (pS409/410; Cosmo Bio, Tokyo, Japan; Cat# TIP-PTD-M01; 1:1000), and BARD1 (E-11; Santa Cruz Biotechnology, Santa Cruz, CA, USA; Cat# sc-74559; 1:1000) for 2 h at room temperature. The sections were subsequently immunostained with biotinylated secondary anti-mouse IgG antibody (Vector Laboratories, Burlingame, CA, USA) and visualized using the avidin-biotin enzyme complex (Vector Laboratories) and 3,3′-diaminobenzidine. We previously validated human BRCA1 detection with this anti-BRCA1 antibody by cellular knockdown studies, and absorption studies using BRCA1 antigens on human brain samples with tau aggregate depositions [7]. We further validated human BRCA1 detection with this antibody by a western blot of the lysate of cells overexpressing human full-length BRCA1 (Figure A1). The human BARD1 detection with the anti-BARD1 antibody used in this study was also validated by cellular overexpression study (Figure A2).

For immunofluorescence double staining, the sections were incubated with 0.1% Triton for 20 min for permeabilization, followed by 20 min of 0.1% Sudan Black/70% ethanol to reduce autofluorescence. The above-mentioned primary antibodies were labeled with Alexa 488 or Alexa 594 using a Zenon Tricolor Mouse IgG1 Labeling Kit (Thermo Fisher Scientific) according to the manufacturer’s instructions. The primary antibody dilution was 1:500 for p-tau and BRCA1 and 1:250 for BARD1. The Zenon Mouse IgG1 labeling reagent was added to obtain a 6:1 molar ratio of Fab to antibody targets. Nuclei were visualized using 4′,6-diamidino-2-phenylindole (DAPI).

Images were captured by an Axioplan 2 fluorescent microscope (Carl Zeiss, Oberkochen, Germany) with an Axiocam HRc Charge-Coupled Device (CCD) camera system (Carl Zeiss).

### 2.3. Extraction of Sarkosyl-Soluble and -Insoluble Fractions

The frozen temporal lobes of 2 AD brain samples, precentral lobes of 2 NC and 3 PSP brain samples, and midbrain of 1 PSP brain sample were used. These brain regions were chosen because of the degree of tau deposition visualized on systemic neuropathological examination. Frozen PiD brain samples were not available. HEK293FT cells transfected with a pcDNA6.2-Venus-BRCA1 plasmid, with or without pcDNA6.2-P301L 1N4R tau plasmids and in vitro generated tau seeds (methods described in the Appendix A section) in 6-well plate were subjected to the same process.

Extraction of the sarkosyl-insoluble fraction from brain samples was performed as previously described [15,16] with minor modifications. Briefly, 150 mg of brain samples was homogenized with a Teflon homogenizer (10 strokes) in 900 µL of homogenization buffer (10 mM Tris-HCl, pH 7.5 containing 0.8 M NaCl, 1 mM ethylene glycol bis(2-aminoethylether)-*N*,*N*,*N*′,*N*′-tetraacetic acid (EGTA), and 1 mM dithiothreitol (DTT)), and 100 µL of 10% sarkosyl (N-lauroylsarcosine) was added to obtain a final sarkosyl concentration of 1%. After 30 min at room temperature, samples were sonicated and ultracentrifuged at 100,000× *g* for 20 min. The supernatant was preserved as the “sarkosyl-soluble fraction,” and the pellet was resuspended in the same buffer containing 1% sarkosyl and washed via ultracentrifugation at 100,000× *g* for 20 min. The pellet was resuspended in 70% formic acid and incubated at 50 °C for 30 min. After evaporation, samples were resuspended in PBS and subjected to sonication to obtain the “sarkosyl-insoluble fraction.”

### 2.4. Western Blot Analyses

Samples were incubated at 65 °C with LDS buffer and 1% 2-mercaptoethanol for 15 min, loaded on 4–15% Mini-PROTEAN TGX gels (Bio-Rad, Hercules, CA, USA), separated by LDS-PAGE, and transferred to a polyvinylidene difluoride (PVDF) membrane with the Trans-Blot Turbo Blotting System and Trans-Blot Turbo Transfer Pack (Bio-Rad, Hercules, CA, USA) using the high molecular weight protocol. Multicolor Protein Ladder (Nippon Gene, Tokyo, Japan) was used as molecular weight markers. Non-specific binding was blocked with EzBlock CAS (ATTO Corporation, Tokyo, Japan) for 1 h at room temperature. Membranes were then incubated with primary antibodies diluted by Can Get Signal Solution 1 (Toyobo, Osaka, Japan) overnight at 4 °C. Anti-BRCA1 (MS110; Abcam; Cat# ab16780; 1:1000), anti-human Tau (HT7; Thermo Fisher Scientific; Cat# MN1000; 1:2000), and anti-actin (C4; Merck Millipore; Cat# MAB1501; 1:4000) antibodies were used as primary antibodies. Membranes were washed with TBST for 5 min three times and subsequently incubated with secondary horseradish peroxidase-conjugated antibodies (GE Healthcare, Little Chalfont, UK)diluted in a 1:2000 ratio in Can Get Signal Solution 2 (Toyobo, Osaka, Japan) for 1 h at room temperature. After being washed with TBST for 5 min three times, membranes were visualized using EzWestLumi Plus, and images were captured by LuminoGraph I (ATTO, Tokyo, Japan). Using the same sample preparation and western blot protocol in HEK293FT cells overexpressing human full-length BRCA1, we determined that in our protocol, human full-length BRCA1 proteins were detected at ≈300 kDa (Figure A1).

### 2.5. Quantification and Statistical Analyses

Western blot of sarkosyl-insoluble fractions of human brain samples were technically quadruplicated. Signal intensity at ≈300 kDa were quantified by Image J software (http://imagej.nih.gov/ij/). Intensity relative to the average intensity of AD samples was used. Statistical analysis was performed with EZR (Saitama Medical Center, Jichi Medical University, Saitama, Japan), which is a graphical user interface for R (The R Foundation for Statistical Computing, Vienna, Austria). Data are summarized by the median and interquartile range (IQR), and the significance of the difference between NC and PSP were determined using the Mann–Whitney U test. A p-value less than 0.05 was considered statistically significant.

## 3. Results

AD patients’ brains showed colocalization of BRCA1 with tau in NFTs and neuropil threads (Figure 1A–D, Figure 2A, and Figure A3), and slight cytoplasmic AT8 positivity observed in the NC patients’ brains in the entorhinal cortex and hippocampal CA1 did not show colocalization of BRCA1 (Figure 1E,F and Figure 2B) as previously described [7,8]. PiD patients’ brains showed colocalization of BRCA1 with tau in Pick bodies in the frontal cortex (Figure 1G,H, Figure 2C and Figure A4). PSP patients’ brains showed colocalization of BRCA1 with tau in globose NFTs found abundant in the midbrain and glial coiled bodies in the cerebral white matter (Figure 1I–L, Figure 2D,E, Figure A5 and Figure A6). Some brain samples showed partial colocalization of BRCA1 with tau in tuft-shaped astrocytes (Figure 1M,N and Figure 2F), while most did not (Figure 1O,P and Figure A7). In CBD patients’ brains, no cytoplasmic BRCA1 staining was observed close to pretangles or astrocytic plaques (Figure 1Q–T). Cytoplasmic BRCA1 positivity in phosphorylated α-synuclein or phosphorylated TDP-43 inclusions was not observed in the DLB, PDD, MSA-C, or ALS brain samples (Figure 3).

Since BRCA1 colocalized with tau aggregates in PSP patients’ brains, we also tested whether BRCA1 aggregated in PSP patients’ brains as we reported in AD patients’ brains [7] by western blot analyses of the sarkosyl-insoluble fraction of brain samples. Although BRCA1 also colocalized with tau aggregates in PiD patients’ brains, frozen brain samples were not available for PiD. Western blot analyses showed insoluble BRCA1 in the AD brain samples and to a lesser extent in the PSP brain samples (Figure 4). Relative BRCA1 intensity at ≈300 kDa were NC = median 0.126 (IQR 0.090–0.159) vs. PSP = median 0.236 (IQR 0.145–0.364) (*p* = 0.0192).

A cellular model was used to see insoluble full-length BRCA1 in the presence of tau aggregation (Methods in Appendix A). The seeding potential of the tau seeds we made was confirmed by a Western blot of the sarkosyl-fractionated samples (Figure A8). Although cells transfected with wild type 4R tau and tau seeds also showed insoluble tau, we used aggregation-prone P301L tau plasmid for further studies since the amount of insoluble tau was much higher (Figure A8).

Western blot of HEK293FT cells overexpressing Venus-tagged human full-length BRCA1 showed that BRCA1 becomes sarkosyl-insoluble only in the presence of tau aggregation by transfection with P301L tau plasmid and in vitro generated tau seeds (Figure 5). 

Since BRCA1 has been reported to form an obligate heterodimer with BRCA1-associated RING domain protein 1 (BARD1) [6] and this heterodimer formation is important for nuclear import [17] and interaction with proteins associated with microtubules and the centrosome [18,19], we hypothesized that BARD1 may be related to the mislocalization of BRCA1 observed in tauopathy patients. Therefore, we also tested BARD1 localization in the AD, PiD, and PSP brain samples. Immunohistochemical studies did not show cytoplasmic BARD1 positivity similar to BRCA1, and immunofluorescence double staining showed that weak BARD1 fluorescence remained in the nucleus and BARD1 did not colocalize with BRCA1 in the cytoplasm (Figure 6).

## 4. Discussion

In this study, we showed that BRCA1 colocalizes with tau aggregates in the cytoplasm in not only AD but also in PiD and PSP. Sarkosyl-insoluble BRCA1 reported in AD brain samples was also observed in PSP brain samples.

While all tauopathies are characterized by aggregated tau protein deposition in the brain, each disease shows different neuropathology and tau strains. NFTs in AD are composed of both 3- and 4-repeat tau isoforms [20], whereas tau filaments in PSP and CBD are composed of only the 4-repeat tau isoform and those in PiD are composed of only the 3-repeat tau isoform [21,22,23]. While AD, PiD, PSP, and CBD all show tau inclusions in neurons, PSP and CBD also show tau inclusions in glial cells; for example, in PSP, glial coiled bodies in oligodendrocytes and tuft-shaped astrocytes in astrocytes [24]. Our results suggest that the coaggregation of BRCA1 and tau can occur in both three-repeat and four-repeat tauopathies and in both neurons and glial cells.

Very recently, Nakamura et al. have also reported BRCA1 colocalization with tau aggregates in human tauopathies [11]. In line with our results, they reported BRCA1 colocalization with tau aggregates in NFTs and neuropil threads in AD, Pick bodies in PiD, and globose NFTs, coiled bodies, and tuft-shaped astrocytes in PSP [11]. They also reported BRCA1 colocalization with tau aggregates in frontotemporal dementia with parkinsonism linked to chromosome 17 (FTDP-17) [11]. In CBD, although BRCA1 colocalization with tau aggregates was not observed in the precentral gyrus and midbrain in all of the three patients in our study, Nakamura et al. reported BRCA1 colocalization with astrocytic plaques and intracytoplasmic inclusions in the basal ganglia (caudate and putamen) in all of the five patients in their study [11]. Whether the difference in the results of CBD patients was due to lesion distribution or other patient characteristics needs further evaluation. Nakamura et al. also reported results using antibodies against BRCA1 phosphorylated at Ser1423 (pBRCA1 (Ser1423)) and at Ser1524 (pBRCA1 (Ser1524)), and showed that while tau aggregates were also positive for pBRCA1 (Ser1423) in all tauopathies, pBRCA1 (Ser1524) was mainly associated with Pick bodies in PiD [11]. 

In our study, although NC and CBD brains also showed p-tau depositions in neurons, pretangles, BRCA1 did not colocalize with these depositions, as reported in NC by Evans et al. [8], and in PSP by Nakamura et al. [11]. Since pretangles are mainly composed of soluble p-tau, and our previous co-immunoprecipitation study using AD brains did not suggest direct binding of BRCA1 to soluble tau [7], we assume that this is most likely because BRCA1 is sequestered in tau aggregates in later phases of aggregation.

In AD and PiD, most of the tau aggregates in neurons were immunopositive for BRCA1 (Figure A3 and Figure A4). In PSP, most of the tau aggregates in neurons (globose NFTs) and oligodendrocytes (glial coiled bodies) were immunopositive for BRCA1 (Figure A5 and Figure A6), while only a small amount of the tau aggregates in astrocytes (tuft-shaped astrocytes) were immunopositive for BRCA1. Differences in the amount of aggregated tau within the cytoplasm between cell types may contribute to this difference as suggested by Nakamura et al. [11], or difference in the expression level or posttranslational modification of BRCA1 between cell types may also contribute as well. Considering these two factors, insolubility of tau and cell types, there seems to be no obvious difference between 3R tau and 4R tau regarding BRCA1 colocalization. 

Our group and other groups have recently reported the importance of BRCA1 in mature neurons [7,8,9,10,25,26,27]. In cancer cells, functions of BRCA1 are most established in homologous recombination, which can only be seen in dividing cells [5,6]. However, BRCA1 has also been reported to be involved in non-homologous end-joining [28] and transcription-associated homologous recombination repair [29,30], which also occur in non-dividing cells, including mature neurons [31,32,33]. Since DNA damage in neurons has been reported in various neurodegenerative diseases [34,35], including PiD [36,37] and PSP [37], the mislocalization and coaggregation of BRCA1 with tau suggested in PiD and PSP in this study may also be involved in the progression of these diseases. Other functions of BRCA1, including regulation of cell cycle [38,39,40] and oxidative stress [27,41], which have also been suggested to be related to neurodegenerative diseases including AD, PiD, and PSP [42,43,44,45,46], may also be involved as suggested by other authors [8,9,11].

While we and others have reported that a decrease in the amount of BRCA1 causes cellular dysfunction in neurons [7,10], whether the decrease of BRCA1, likely caused by coaggregation with tau in PSP patients’ oligodendrocytes, has pathological meanings needs further investigation. Since DNA damage and the importance of DNA repair proteins have been suggested as well in oligodendrocytes [47], a decrease in the amount of functional BRCA1 in oligodendrocytes may also cause cellular dysfunction.

Although previous cancer research showed that the majority of BRCA1 proteins form a heterodimer with BARD1, BARD1 did not colocalize with BRCA1 in tau aggregates in the AD, PiD, and PSP brain samples. Nakamura et al. have also reported that the BARD1 antibody they used did not react with tau inclusions in tauopathy patients’ brains [11]. Since heterodimer formation with BARD1 is important for nuclear localization of BRCA1, one hypothesis is that BRCA1 dissociated from BARD1 during export from the nucleus and subsequently coaggregated with tau in the cytoplasm. Another hypothesis is that BRCA1 dissociated from BARD1 during coaggregation with tau in the cytoplasm.

There are several limitations to this study. First, since the sample size was relatively small and all patients were Japanese, more studies are needed to determine whether our findings apply to other PiD, PSP, and CBD patients. Second, future studies are needed to determine whether BRCA1 also colocalizes with tau in other tauopathies, such as argyrophilic grain disease, chronic traumatic encephalopathy, tangle-only dementia, amyotrophic lateral sclerosis, and parkinsonism–dementia complex, and globular glial tauopathy [1]. Third, the reason why tau aggregates in certain cell types seem to be more susceptible to BRCA1 colocalization needs further investigation.

## 5. Conclusions

BRCA1 was mislocalized to the cytoplasm and colocalized with tau aggregates in not only the AD brain samples but also in the PiD and PSP brain samples. Coaggregation of BRCA1 with tau may also be involved in the pathogenesis of PiD and PSP. 

## Figures and Tables

**Figure 1 brainsci-10-00007-f001:**
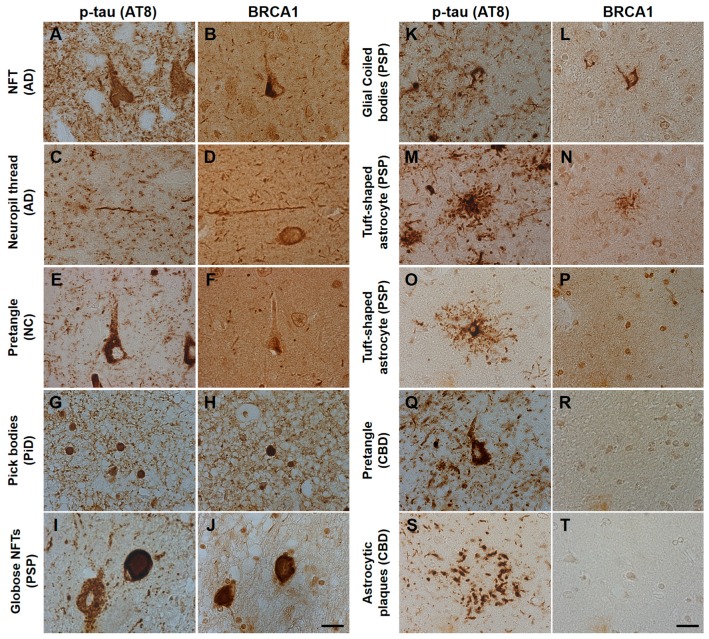
Immunohistochemical studies using antibodies against phosphorylated tau (AT8) and BRCA1 (MS110) in human tauopathy patients’ brains. (**A**–**D**) Neurofibrillary tangles (NFTs) and neuropil threads in Alzheimer’s disease (AD) patients’ brains were immunopositive for BRCA1 (**E**,**F**). In normal control (NC), BRCA1 positivity was observed in the nucleus in the hippocampus despite AT8 positivity in the cytoplasm. (**G**,**H**) Pick bodies in Pick’s disease (PiD) were immunopositive for BRCA1. (**I**–**L**) Globose NFT in the midbrain and precentral cortex and glial coiled bodies in the cerebral white matter of progressive supranuclear palsy (PSP) patients’ brains were immunopositive for BRCA1. (**M**–**P**) Although some BRCA1 positivity partially resembling tuft-shaped astrocytes was observed in the frontal cortex (**M**,**N**), cytoplasmic BRCA1 positivity was not observed close to most of the AT8-positive tuft-shaped astrocytes in PSP patients (**O**,**P**). (**Q–T**) In corticobasal degeneration (CBD) patients’ brains, no cytoplasmic BRCA1 staining was observed close to pretangles or astrocytic plaques. p-tau, phosphorylated tau. The scale bars represent 20 µm.

**Figure 2 brainsci-10-00007-f002:**
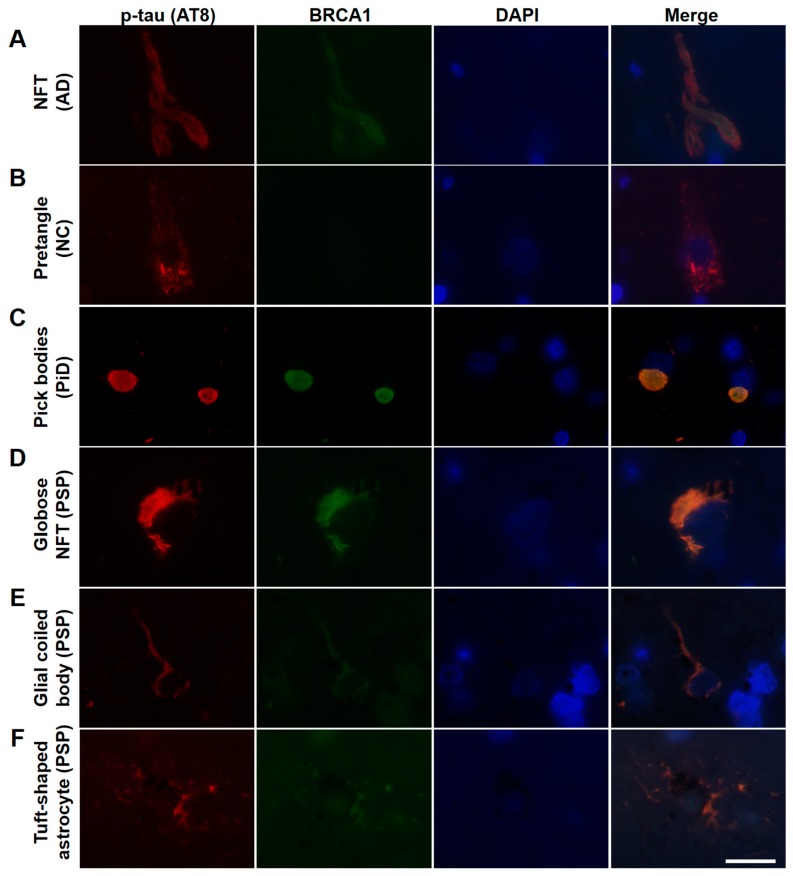
Immunofluorescence double staining of human tauopathy patients’ brains using antibodies against phosphorylated tau (AT8) and BRCA1 (MS110). Colocalization of p-tau and BRCA1 was observed in NFTs of the AD brain samples (**A**), Pick bodies of the PiD brain samples (**C**), and globose NFTs (**D**), glial coiled bodies (**E**), and some of the tuft-shaped astrocytes in the PSP brain samples (**F**) but not in the NC brain samples (**B**). The scale bars represent 20 µm.

**Figure 3 brainsci-10-00007-f003:**
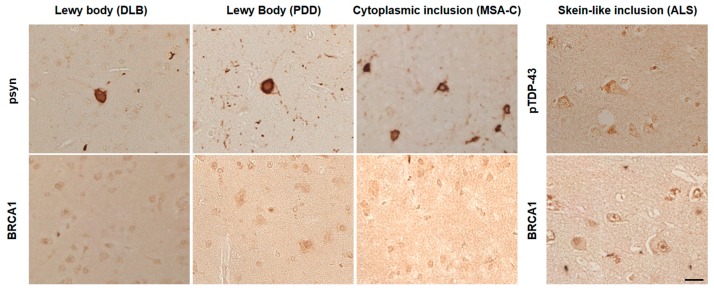
Immunohistochemical studies using anti-BRCA1 antibodies in disease controls. BRCA1 positivity resembling phosphorylated α-synuclein (psyn) or phosphorylated TDP-43 (pTDP-43) positive inclusions was not observed in dementia with Lewy bodies (DLB), Parkinson’s disease with dementia (PDD), multiple system atrophy cerebellar type (MSA-C), or amyotrophic lateral sclerosis (ALS). The scale bar represents 20 μm.

**Figure 4 brainsci-10-00007-f004:**
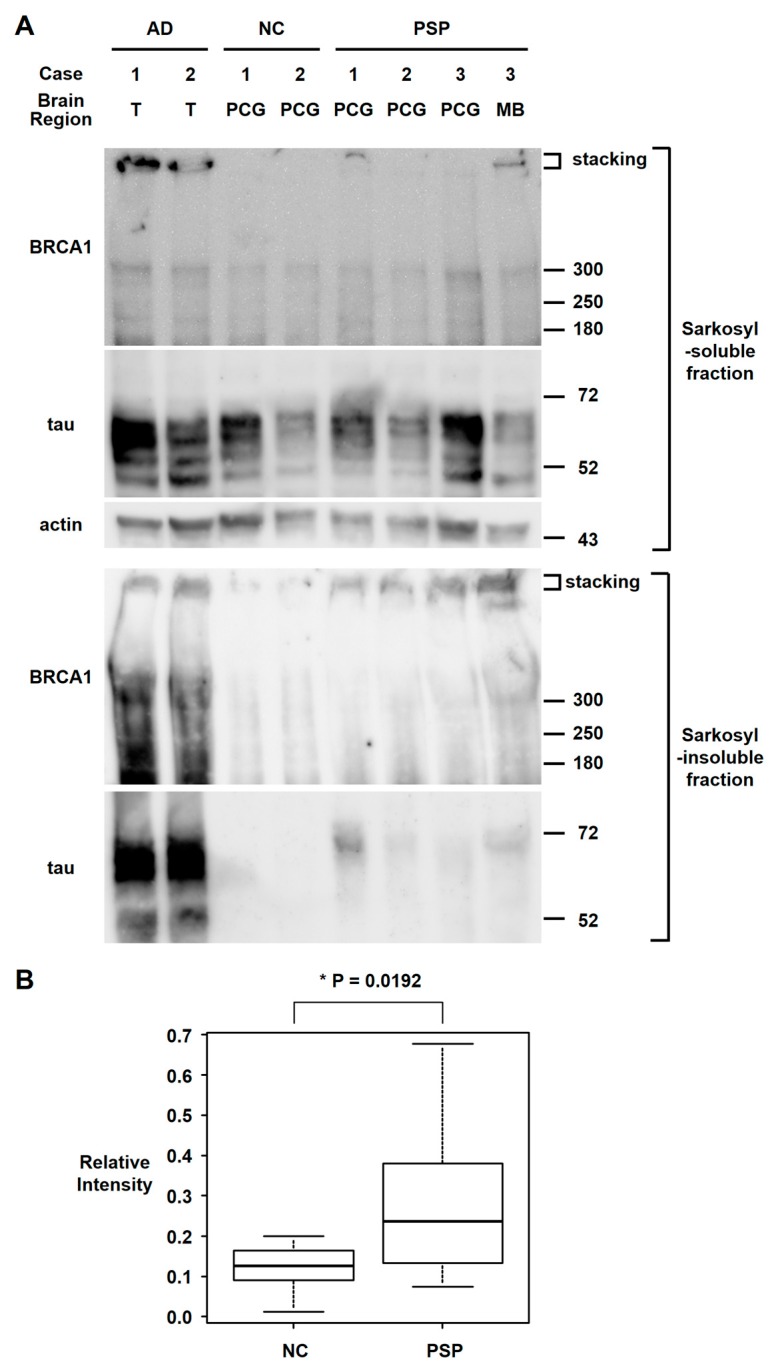
Western blot analyses of the sarkosyl fractionated human brain samples. (**A**) Sarkosyl-soluble fraction of brain samples showed similar intensity bands of full-length BRCA1 (≈300 kDa) among the AD, NC, and PSP brain samples, with high intensity stacking observed in 2 AD brain samples and 1 PSP midbrain sample. In the sarkosyl-insoluble fraction, tau bands were only observed in AD and PSP brain samples. Sarkosyl-insoluble full-length BRCA1 (≈300 kDa) was observed in both AD samples as previously described. Although sarkosyl-insoluble full-length BRCA1 monomer band intensities in PSP samples were higher than in the NC in only 1 midbrain sample, the stacking intensities were higher in PSP samples, which suggests BRCA1 aggregation. T: temporal cortex, PCG: precentral gyrus, MB: midbrain. (**B**) Quantification of sarkosyl-insoluble BRCA1 signal at ≈300 kDa. Intensity relative to the average intensity of AD samples is shown in box and whisker plots. Although much less than AD, PSP brain samples showed significantly more insoluble BRCA1 than NC.

**Figure 5 brainsci-10-00007-f005:**
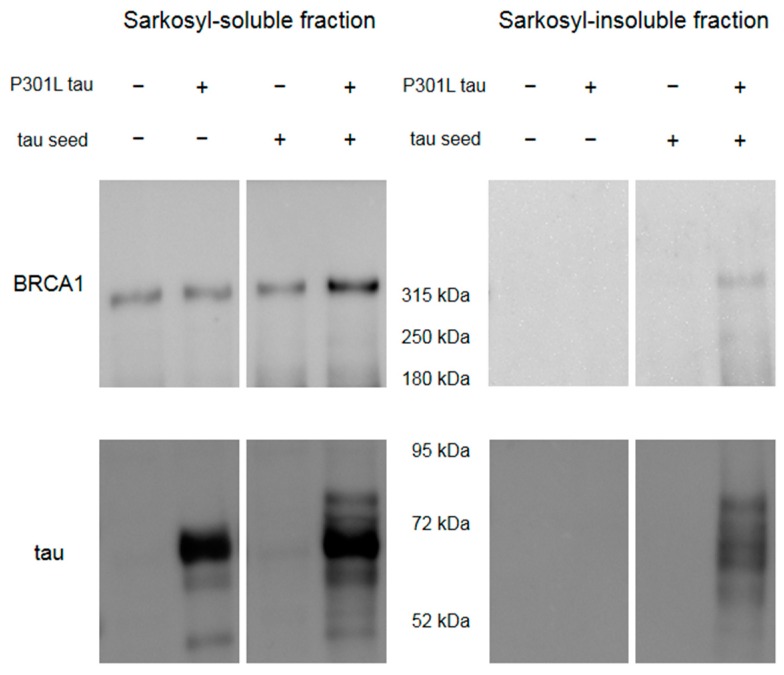
Western blot of HEK293FT cells overexpressing Venus-tagged human full-length BRCA1 with or without transfection with P301L tau and in vitro generated tau seeds. Sarkosyl-insoluble BRCA1 was observed only in the cell sample transfected with both P301L tau plasmid and tau seeds, which showed sarkosyl-insoluble tau.

**Figure 6 brainsci-10-00007-f006:**
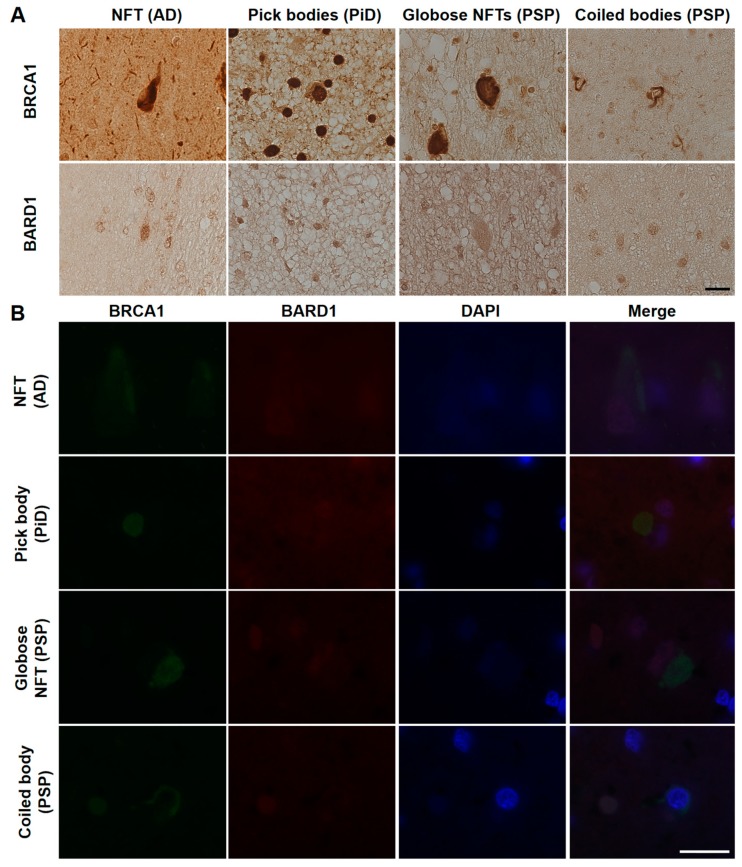
Immunohistochemical (**A**) and immunofluorescence double staining (**B**) studies using antibodies against BRCA1 and BARD1 in human tauopathies. (**A**) Cytoplasmic BARD1 positivity resembling tau aggregates was not observed in AD, PiD, or PSP brain samples. (**B**) BARD1 did not colocalize with BRCA1 in the cytoplasm and showed weak positivity in the nucleus. NFT, neurofibrillary tangle. The scale bars represent 20 µm in both A and B.

**Table 1 brainsci-10-00007-t001:** Clinical and neuropathological characteristics of the studied patients.

Case	Pathological Diagnosis	Age (Years)	Sex	PMI (h:min)	BW (g)	CDR	Aβ (Braak)	NFT (Braak)	Lewy (BBAR)
1	AD	78	M	5:36	1444	2	C	V	1
2	AD	86	F	8:18	1010	3	C	V	0.5
3	AD	77	M	11:25	1310	2	C	V	2
4	AD	86	F	20:27	1146	2	C	V	0.5
5	PiD	74	M	7:12	1048	3	A	I	0
6	PiD	74	F	5:24	730	3	A	II	0
7	PSP	80	M	4:03	1178	2	A	I	0
8	PSP	84	M	17:44	1510	3	A	III	0
9	PSP	75	M	9:07	1340	3	C	II	0
10	CBD	79	M	12:16	1266	3	0	II	0
11	CBD	74	F	53:30	899	3	A/B	I	0
12	CBD	69	F	3:20	819	3	A	I	0
13	NC	80	F	18:31	1280	0	0	II	0
14	NC	84	F	14:37	1030	0	A	I	1
15	NC	83	F	17:43	1440	N/A	0	I	0
16	NC	79	M	11:30	1166	0	A	II/I	0
17	DLB	88	M	1:40	1055	1-2	0	I	4
18	PDD	85	M	14:42	1440	3	0	II	4
19	MSA-C	70	M	3:00	1240	N/A	N/A	N/A	N/A
20	ALS	75	F	3:35	1170	N/A	N/A	N/A	N/A

PMI, post mortem interval; BW, brain weight; CDR, Clinical Dementia Rating; Aβ (Braak), Braak staging of amyloid β deposits; NFT (Braak), Braak staging of neurofibrillary tangles; Lewy (BBAR), the Brain Bank for Aging Research staging for Lewy body pathology; AD, Alzheimer’s disease; PiD, Pick’s disease; PSP, progressive supranuclear palsy; CBD, corticobasal degeneration; NC, normal control; DLB, dementia with Lewy bodies; PDD, Parkinson’s disease with dementia; MSA-C, multiple system atrophy cerebellar type; ALS, amyotrophic lateral sclerosis; M, male; F, female; N/A, not available.

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
