# Peer review of "Colocalization of BRCA1 with Tau Aggregates in Human Tauopathies"

_brainsci, 2019, doi:10.3390/brainsci10010007_

Round 1

Reviewer 1 Report

The Authors present important in the field aspects of the significance of oxidative-stress driven damages occurring during neurodegeneration. Precisely, they continue their work over BRCA1 protein, one of the major player of oxidative DNA damage response, in Alzheimer’s disease. In this work, they extended their study to other neurodegenerative disorders and they indicate that BRCA1 was mislocalized to the cytoplasm and colocalized with tau aggregates in not only the AD brain samples but also in the PiD and PSP brain samples. The authors used valuable postmortem brains material. However, such material corresponds to the terminal stage of the disease. Thus, such research should be continued using the cellular in vitro models, likely addressing less advanced disease stages.

To provide the full landscape on the role of BRCA1 in AD some citations should be added. The citations should be incorporated in the introduction section somewhere around the fragment: “In AD, although DNA damage induced by extracellular amyloid β (Aβ) was accompanied by upregulation of BRCA1 protein in neurons, BRCA1 aggregated with tau in the cytoplasm [7]. In AD mouse models, knockdown of BRCA1 in the hippocampus in the presence of extracellular Aβ increased the most toxic form of DNA damage, DNA double-strand breaks, and impaired the neuronal morphology, electrophysiological characteristics, and cognitive function in mice [7, 8]. Thus, coaggregation of BRCA1 with tau observed in advanced AD patients and reduced functional BRCA1 in the nucleus can lead to insufficient DNA repair and neuronal dysfunction.”. The citations should be also added in the discussion (in the sentence: “Our group and other groups have recently reported the importance of BRCA1 in mature 232 neurons [7, 8, 23–25].”). The followed data should be cited:

Evans TA, Raina AK, Delacourte A, Aprelikova O, Lee HG, Zhu X, Perry G, Smith MA. BRCA1 may modulate neuronal cell cycle re-entry in Alzheimer disease. Int J Med Sci. 2007 May 12;4(3):140-5.

Key information: Correlation between upregulated and AD was already described by Evans et al. in 2007. They found that BRCA1 is upregulated and relocalized to the cytosol in post-mortem brains derived from late-onset SAD patients. The studies of Evans et al. contributed to the cell cycle re-entry hypothesis in Alzheimer disease and indicated that the neurofibrillary tangles in AD may have a different genesis from those in normal aging.

Wezyk M, Szybinska A, Wojsiat J, Szczerba M, Day K, Ronnholm H, Kele M, Berdynski M, Peplonska B, Fichna JP, Ilkowski J, Styczynska M, Barczak A, Zboch M, Filipek-Gliszczynska A, Bojakowski K, Skrzypczak M, Ginalski K, Kabza M, Makalowska I, Barcikowska-Kotowicz M, Wojda U, et al. Overactive BRCA1 Affects Presenilin 1 in iPSC-Derived Neurons in Alzheimer's Disease. J Alzheimers Dis. 2018;62(1):175-202.

Key information: Another group has also recently indicated upregulation of re-localized BRCA1 in AD (Wezyk et al. 2018). They showed increased levels of BRCA1 phosphorylated at Ser1524 and re-localized to the cytosol in fibroblasts, neural stem cells (NSCs) and induced pluripotent stem cell (iPSC)-derived neurons obtained from familial AD (FAD) patients. Accordingly, they showed significant upregulation of BRCA1-Δex11 in AD cell lines, an isoform specific for cytosol localization. They also found that BRCA1(Ser1524) colocalized with presenilin 1, a major component of the γ-secretase complex, which further contributes to abnormal amyloid-beta 42 (Aβ42) peptide production, in fibroblasts, as well as with amyloid-beta in neurons. They suggested that BRCA1 may affect the turnover of presenilin 1 (PS1), and thus amyloidogenesis. Finally, they observed that increased BRCA1(Ser1524) was followed by recruitment of the Cdc25C phosphatase phosphorylated at Ser216 in AD neurons, suggesting activation of the cell cycle re-entry in postmitotic neuronal cells.

Several questions/comments on the results and discussion sections:

Can author tell what types of neurons are affected? Based on their morphology? Based on typical structural abnormalities? Or perhaps based on stainings? Nuclei of most BRCA1/p-tau positive cells are enlarged, except pick bodies, as a hallmark of their “dying” condition, it could be commented in the text why pick bodies are not. Also, explain how the cells are confirmed to be a neuronal identity? glial identity? Try to explain more why Pick’s disease and progressive supranuclear palsy are BRCA1-positive, but not other tested diseases, except AD, already previously studied (Mano et al. 2018)? Try to explain what might be the implication of the fact that coaggregation of BRCA1 and tau can occur in both neurons and glial cells, what it could mean to glial? What kind of glia they are? I guess that 3-repeat and 4-repeat tau were distinguishable in the study, so were there implications of different tau isoforms to BRCA1 accumulation /colocalization/aggregation rate or generally to the altered BRCA1 behaviour/activity? (see Chen et al. MAPT isoforms: differential transcriptional profiles related to 3R and 4R splice variants. J Alzheimers Dis. 2010;22(4):1313-29. doi: 10.3233/JAD-2010-101155 or Sealey et al. Distinct phenotypes of three-repeat and four-repeat human tau in a transgenic model of tauopathy. Neurobiol Dis. 2017 Sep;105:74-83. doi: 10.1016/j.nbd.2017.05.003, or other similar reports).

In the discussion the followed sentence in unclear, or not in a good shape: “Third, the reason tau inclusions in the CBD brain samples and most of the tuft-shaped astrocytes in the PSP brain samples did not show colocalization of BRCA1 but other tau inclusions did remains unclear.”

In the sentence: “Differences in tau aggregates or properties of BRCA1 within the cells (e.g., expression level, posttranslational modification) may contribute to these differences. “ I would add that the authors did not test phosphorylated forms or varying functionally isoforms of BRCA1. The research design might include the stainings against some active forms of BRCA1, some post-translationally modified forms, if possible (if specimens are still available, it would be interesting to see the staining for BRCA1 phosphorylated forms, e.g. 1524Ser, 1423Ser – related to apoptosis, or other related to cycle phases transitions or in case of neurons cell cycle re-entry, i.e. mitotic entry, refer to “BRCA1 phosphorylation: Biological consequences” by T. Ouchi 2006, table 1)

Quality of the immunofluorescence stainings / microscopic images can be improved, especially regarding the background.

Comment to the method, i.e. to the description on 3R or 4R tau isoforms stainings using specific primary antibodies. I could not allocate the distinction between 3R and 4R stainings in the data or in table 1. Are they distinguishable?

Author Response

Thank you very much for your valuable comments. Please see the attachment for our responses.

Reviewer 2 Report

The authors have systematically mapped the colocalization of the BRCA1 with Tau aggregates in several human tauopathies complementing their previous study on colocalization of BRCA1 with tau aggregates in Alzheimer’s disease. The study has been appropriately conducted with necessary controls. Though may be out of the scope of this manuscript, it would be interesting to see if BRCA1 colocalizes with other aggregate forms of tau like low molecular weight oligomers.

Author Response

(The authors gave the same response as above.)

Reviewer 3 Report

This paper is a good logical continuation of the previous observations by the group of Dr. Iwata, where they describe for the first time the co-localization of Tau and BRCA1 in Alzheimer disease, among other important findings (Proc Natl Acad Sci U S A. 2017 Nov 7;114(45):E9645-E9654. doi: 10.1073/pnas.1707151114. Epub 2017 Oct 17.). In this paper, the group shows how Tau tangles are positive for BRCA1 in human brains with other tauopathies, such as Pick´s disease or PSP, while it does not co-localize with other aggregation prone proteins such as synuclein or TDP-43. The paper is short and simple, but straight to the point, and the relationship between neurodegeneration and cell proliferation/cancer has been always very intriguing. Unfortunately, this contribution could have been more novel if it wasn´t because other group has just published in October 2019 almost exactly the same results but much more complete (including more samples and quantifications, better images and a more complete view of the current state of the field). The paper in question is this:

Aberrant Accumulation of BRCA1 in Alzheimer Disease and Other Tauopathies.

Nakamura M, Kaneko S, Dickson DW, Kusaka H.

J Neuropathol Exp Neurol. 2019 Oct 19. pii: nlz107. doi: 10.1093/jnen/nlz107.

This paper is not mentioned in the introduction nor the discussion of the current article, but it clearly deprives all novelty to the present paper by Iwata´s group. Iwata´s group should consider the possibility of provide more experiments for their article in order to add novelty. For example, looking for co-localization of other DNA repair proteins in Tau tangles, or even other aggregation-prone proteins, in the same samples. Otherwise, Brain Sciences would be publishing a confirmation paper, and not something really novel. And in the case the editor passes it, I think it would be mandatory to -at the very least- acknowledge the first people to describe these results. But given the extensive work done by the competitors, I would also ask for quantification data, increased sample size, or even complementary  experiments in cell lines, as the authors suggest in their discussion of the limitations of the study.

Also, the attempts to confirm Tau/BRCA1 co-aggregation by immunoblotting of sarcosyl-soluble protein extracts are not very definitive (Figure 4). Although the authors hypothesize that the BRCA1 protein becomes insoluble because there is more signal at the level of the stacking gels (therefore suggesting that the size is so big it does not go through the gel beyond the stacking gel), the fact is that there is no signal where it is supposed to be, as it happens in AD. Further attempts should be made to confirm co-aggregation of tau and BRCA1 by biochemical methods. For example: co-immunoprecipitation studies, filter trap assays, sucrose gradients or the same assays described in this paper but with other detergents, such as triton-X, SDS, ... or alternative conditions.

In case the authors consider revision, I think the article should be completed or expanded significantly with novel results and/or a more intensive and extensive analysis, and of course the paper by Nakamura et al. cited and acknowledged properly.

Author Response

Thank you very much for your valuable comments. We tried our best to improve our manuscript. Please see the attachment for our responses and revisions. 

Round 2

Reviewer 3 Report

The authors have made sufficient attempts to address my concerns, but I think the Nakamura paper should be also acknowledged in the introduction. In lines 57 and 58, the authors state that it is unknown whether BRCA1 and Tau interact in other tauopathies and this is no longer true. Please, rephrase these last sentences so that they adjust better to what we know now.

I still think the authors are publishing virtually the same paper as Nakamura et al, only less complete。 On the other hand, it is good to have independent confirmation of facts, and the authors have a solid line of research on this subject.

In line 56, there is an unfinished sentence: "Others have suggested"

Author Response

(The authors gave the same response as above.)
